# Network oscillation rules imposed by species-specific electrical coupling

**Stefanos Stagkourakis†, Carolina Thörn Pérez†, Arash Hellysaz, Rachida Ammari, Christian Broberger\***

Department of Neuroscience, Karolinska Institutet, Stockholm, Sweden

**Abstract** Electrical junctions are widespread within the mammalian CNS. Yet, their role in organizing neuronal ensemble activity remains incompletely understood. Here, in a functionally well-characterized system – neuroendocrine tuberoinfundibular dopamine (TIDA) neurons - we demonstrate a striking species difference in network behavior: rat TIDA cells discharge in highly stereotyped, robust, synchronized slow oscillations, whereas mouse oscillations are faster, flexible and show substantial cell-to-cell variability. We show that these distinct operational modes are explained by the presence of strong TIDA-TIDA gap junction coupling in the rat, and its complete absence in the mouse. Both species, however, encompass a similar heterogeneous range of intrinsic resonance frequencies, suggesting similar network building blocks. We demonstrate that gap junctions select and impose the slow network rhythm. These data identify a role for electrical junctions in determining oscillation frequency and show how related species can rely on distinct network strategies to accomplish adaptive control of hormone release.
DOI: https://doi.org/10.7554/eLife.33144.001

## Introduction

The default activity pattern of a neuronal network emerges from the combination of the intrinsic electrical characteristics of the constituent neurons and the wiring and properties of cell-to-cell connections within the circuit. In order to understand the operations of the brain, it is therefore necessary to elucidate the precise contribution of membrane electrophysiology, as well as different modes of communication, to ensemble activity. Neuronal gap junctions (*Furshpan and Potter, 1957*; *Watanabe, 1958*; *Bennett et al., 1963*) impact on both these features. While once surmised to be exclusive to invertebrate species and/or early development, steadily accumulating evidence has established that electrical junctions formed by gap junctions are both common and consequential within many populations in the adult, mammalian CNS (see *Connors and Long, 2004*). By mediating ultra-fast, subthreshold, bidirectional (albeit not always symmetrical) cell-cell communication, gap junctions have been implicated most obviously in the synchronization (e.g. *Christie et al., 1989*; *Beierlein et al., 2000*; *Dugué et al., 2009*) of neuronal activity. The particular kinetics of electrical junctions further allow them to for example interact with the parallel chemical synaptic transmission between neurons (e.g. *Llinas et al., 1974*; *Mathy et al., 2014*; see *Pereda, 2014*) and spread intracellular messengers between cells (*Tsien and Weingart, 1974*). The presence of gap junctions changes the computational rules within circuits.

Gap junctions are of particular interest in the context of neuronal oscillations, a collective behavior that involves the coordination, and often harmonization, of neuronal discharge (e.g. *Manor et al., 1997b*; *Draguhn et al., 1998*; *Hormuzdi et al., 2001*). Oscillations are intrinsic to brain networks throughout the neuraxis (see *Buzsáki and Watson, 2012*), appearing in wide, but stereotyped, frequency spectra. Shifts in network frequency have been proposed to correspond to changes in the information carried across neuronal transmission (see *Singer and Gray, 1995*), and a switch in period can denote the conversion of a physiological oscillation into a pathological rhythm

**\*For correspondence:**
Christian.Broberger@ki.se

†These authors contributed equally to this work

**Competing interests:** The authors declare that no competing interests exist.

(*Kostopoulos et al., 1981*). While the issue of whether (and if so, how) gap junctions influence, or even set, the specific frequency of a network oscillation has been the subject of a limited number of elegant modeling and theoretical studies (*Kepler et al., 1990*; *Meunier, 1992*; *Pedersen et al., 2005*), there is little experimental data addressing this important issue. This dearth of information can in no small measure be ascribed to limitations in the available methodological repertoire: pharmacological tools are compromised by poor specificity (*e.g. Beaumont and Maccaferri, 2011*), whereas genetically manipulated animals, although informative in several aspects, often exhibit incomplete loss-of-function (*De Zeeuw et al., 2003*; *Lee et al., 2014*). Additional models could thus be of substantial value to forward our understanding. Here, studying the neuroendocrine tuberoinfundibular dopamine (TIDA) neurons, which oscillate between depolarized UP and hyperpolarized DOWN states (*Lyons et al., 2010*), we uncover an unexpected rat-mouse species difference in electrical coupling that we exploit to reveal principles of how gap junctions can constrain and dictate ensemble network behavior.

## Results

Tuberoinfundibular dopamine neurons in the hypothalamic dorsomedial arcuate nucleus (dmArc) of rats exhibit a slow, robust, highly rhythmic, synchronized network oscillation in vitro (*Lyons et al., 2010*). Reports on mouse TIDA neurons, however, describe notably irregular fast phasic firing (*Brown et al., 2012*; *Romanò et al., 2013*; *Zhang and van den Pol, 2015*). Such dichotomy across analogous rodent circuits are rare in the literature. This intriguing observation prompted us to systematically compare rat and mouse TIDA neurons under identical recording conditions to first determine if these discordant accounts reflect a species difference proper or discrepancies in methodology.

Male rat TIDA cells identified by their location in the dmArc (*Figure 1A–B*) and characteristic electrical properties (*Lyons et al., 2010*), exhibited a stereotyped slow (0.15 ± 0.01 Hz; n = 29, *Figure 1C,G*) and highly regular (coefficient of variation (CV) for frequency = 1.99 ± 0.69%; n = 8 slices; *Figure 1H*) oscillation with similar membrane voltage across neurons within slices and between animals (*Lyons et al., 2010*) (*Figure 2A,B,D*). In striking contrast, while age- and sex-matched mouse TIDA neurons (*Figure 1D,E*), identified by using transgenic dopamine transporter (DAT; encoded by the *Slc6a3* gene)-tdTomato–expressing mice (*Figure 1E*), also exhibited sub - or supra-threshold phasic firing across different experimental conditions (*Figure 1F*, n = 284/311; *Figure 2B–E*), these oscillations were significantly faster (0.39 ± 0.05 Hz; n = 18–29 per group, *Figure 1F,G*), less rhythmic (*Figure 1H*), of a wide range of frequencies within the same slice (*Figure 1G*; CV = 41.48 ± 6.19%; n = 8 slices) and more depolarized (mouse oscillation nadir −51.2 ± 0.9 mV *vs.* rat −66.0 ± 1.1 mV; n = 18 per group).

In addition to these temporal discrepancies, oscillation robustness varied markedly between species. An oscillatory current was seen in voltage clamp mode only in rat TIDA neurons (*Figure 1C,F*). Furthermore, rat TIDA oscillation frequency was insensitive to hyperpolarization (*Figure 1I–K*), whereas mouse TIDA cells progressively decreased their frequency in response to negative current until it collapsed to a stable membrane potential below *ca.* −80 mV (*Figure 1L–N*). To evaluate entrainability, we used the sinusoidal ('ZAP'; *Hutcheon and Yarom, 2000b*) current command of increasing frequency. Rat TIDA oscillations maintained their frequency when challenged with ZAP *I* (*Figure 1O,P*). In contrast, application of the same protocol to mouse TIDA neurons resulted in a full entrainment of the cells' rhythm (*Figure 1Q,R*). The cross-correlation coefficient of the frequency of the recorded cell and the current command was thus significantly lower in rat (0.61 ± 0.03; n = 10) compared to mouse (0.87 ± 0.01; n = 10; *Figure 1S,T*), demonstrating the rat TIDA oscillation is robust against perturbation, in contrast to the mouse.

We next addressed if the rat-mouse differences in single-cell features are reflected at the network level. Paired recordings from rat TIDA neurons revealed pronounced phase-locked synchronicity between cells (n = 63 pairs; *Figure 3A,B*; see also *Lyons et al., 2010*) in contrast to mouse TIDA cells, which were asynchronous (n = 40 pairs; *Figure 3C,D*). To record TIDA activity at the population level we performed two-photon Ca$^{2+}$ imaging. Using Oregon green-BAPTA in rat slices, a cluster of cells in the dmArc exhibited spontaneous, slow synchronized Ca$^{2+}$ oscillations (0.15 ± 0.01 Hz; *Figure 3E,F,I*). Ca$^{2+}$ fluxes were visualized in murine TIDA cells using DAT-GCaMP3 mice. Simultaneous whole-cell and Ca$^{2+}$ recordings in mouse TIDA neurons revealed strongly correlated

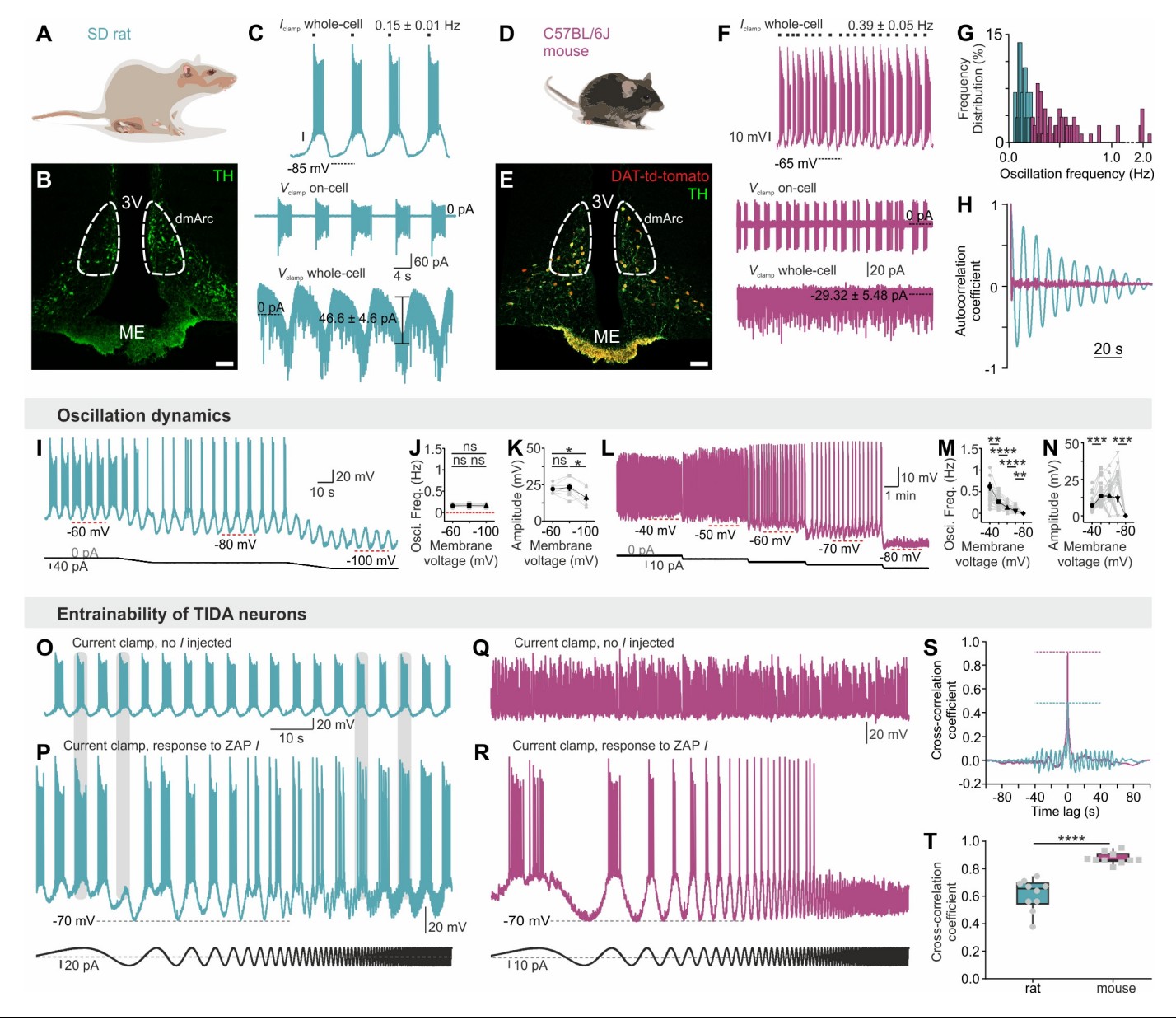

**Figure 1.** Slow, regular, fixed *vs* fast, irregular, entrainable activity in TIDA neurons between two rodent species. (A–B) Distribution of TH-immunostained TIDA neurons (green) in the rat dorsomedial arcuate nucleus (dmArc; coronal section). ME, median eminence; 3V, third ventricle. (C) Patch clamp recordings from a rat TIDA neuron show a slow and robust oscillation evident in whole-cell current clamp (top), on-cell (middle) as well as voltage clamp (bottom) configurations. (D–E) Mouse TIDA neurons in the dmArc visualized by dopamine transporter (DAT)-driven tdTomato fluorescence (red) and TH immunostaining (green). (F) Whole-cell current clamp recording (top) from a mouse TIDA neuron shows fast and irregular phasic firing, also evident in on-cell configuration (middle), whereas no oscillatory current is seen voltage-clamp mode (bottom). (G) Frequency distribution of the oscillation frequency in rat (blue) *vs* mouse (purple) TIDA neurons, with a narrow range of slow frequencies in rat and a wide range of faster oscillation frequencies in mouse (n = 29–115 per group). (H), Representative autocorrelograms of rat and mouse TIDA oscillation. (I) Whole-cell current clamp recording: Progressive hyperpolarization fails to alter oscillation frequency in rat TIDA cells. (J, K,) Quantification of rat TIDA oscillation frequency (J) and amplitude (K) at different voltages (−60 mV, −80 mV and −100 mV; n = 7, one-way ANOVA with Bonferroni as *post-hoc* test). (L) Progressive hyperpolarization (as in I) gradually slows oscillation frequency in mouse TIDA cells, with ultimate collapse of the oscillation below −80 mV. (M, N) Quantification of mouse TIDA oscillation frequency (M) and amplitude (N) at different voltages (n = 34, one-way ANOVA with Bonferroni as post-hoc test). (O–T) Test for entrainability of TIDA neurons in rat vs mouse arcuate slices during whole-cell current clamp recording. (O) Rat TIDA neuron immediately prior to ZAP current (*I*; sinusoidal current of gradually increasing frequency) injection presented in (P). (P) Bottom trace; injection of ZAP *I* does not change the rat TIDA neuron's oscillation frequency. (Q) Mouse TIDA neuron immediately prior to ZAP *I* injection presented in (R). (R) Injection of ZAP *I* entrains mouse TIDA neurons. (S) Cross correlation coefficient example and T) quantification between the injected ZAP *I* and the membrane

*Figure 1 continued on next page*

*Figure 1 continued*

voltage response in the two species (n = 10 per group, p<0.0001, unpaired *t*-test; rat shown in blue, mouse shown in purple). Data expressed as mean ±s.e.m.

DOI: https://doi.org/10.7554/eLife.33144.002

fluctuations in membrane potential and $Ca^{2+}$ activity, indicating that they reflect the same underlying cellular phenomena (*Figure 3—figure supplement 1*). In agreement with electrophysiology, cells exhibited asynchronous, fast (0.40 ± 0.05 Hz; *Figure 3G–I*) oscillation frequencies, with large variability. These data show that the species differences found at the single-cell level are paralleled in the rat and mouse TIDA network.

We next sought to identify the mechanism behind these differences. Network behavior emerges from the intrinsic properties of neurons and their connectivity schemes. In whole-cell recordings, rat and mouse TIDA neurons were found to have similar electrophysiological signatures, such as a prominent A-like $K^+$ current, anomalous inward rectification and slow after hyperpolarization (*Lyons et al., 2010*; *Zhang and van den Pol, 2015*) - *Figure 4A,B,E,F*), albeit of partly different amplitude (*Figure 4I–K*). In both species, oscillations were abolished in the presence of tetrodotoxin (TTX; *Figure 4L*), suggesting a similar dependence on the persistent $Na^+$ current for phasic firing (*Lyons et al., 2010*). Cytoarchitectonic comparison of the TIDA somata size, dendritic lengths and branching (evaluated using Sholl analysis) revealed no differences (*Figure 4—figure supplement 1*). These results indicate that a difference in chemical synaptic or electrical coupling is likely to underlie the two distinct patterns of electrical activity. Importantly, in both rats and mice, oscillations persisted during blockade of fast ionotropic glutamate and GABA transmission (*Figure 4M*), suggesting that chemical synapses are not required for the expression of rhythmic behavior. Our attention therefore turned to electrical junctions (see *Pereda, 2015*; *Connors, 2017*) and their potential role

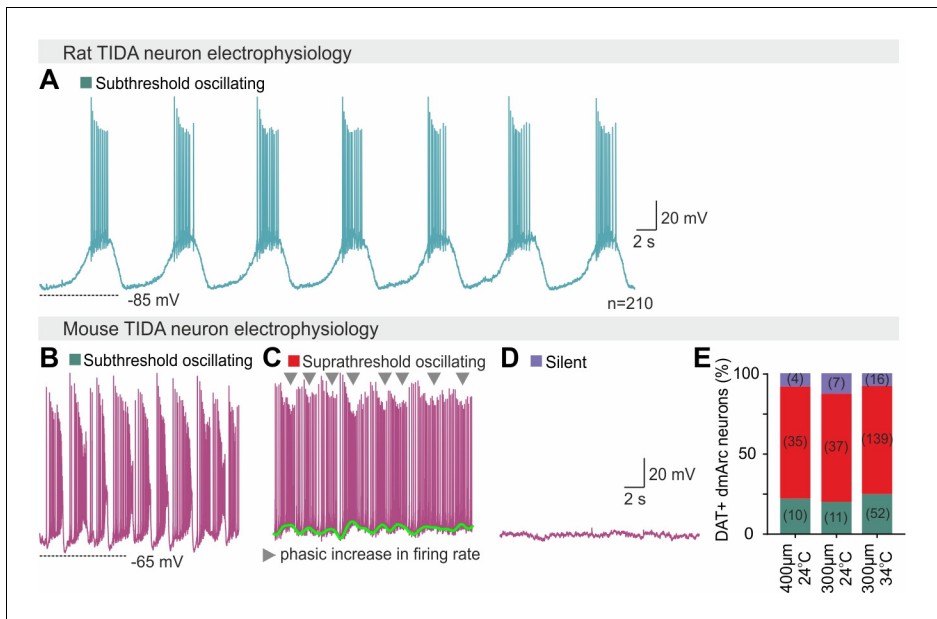

**Figure 2.** Homogeneous vs heterogeneous electrophysiological activity in rat vs mouse TIDA neurons. (**A**) Representative whole-cell current clamp recording of a TIDA neuron from a hypothalamic rat brain slice. (**B–D**) Representative whole-cell current clamp recordings of three mouse TIDA neurons, illustrate the wide spectrum of electrical activity (subthreshold oscillating, suprathreshold oscillating and silent) found in the murine TIDA population. (**E**) Quantification of the percentage of neurons found at the three different states shown in (**b–d**) under different recording conditions. Note that the proportions remain the same regardless of network volume (slice thickness, 300 or 400 μm) or temperature (24°C or 34°C).

DOI: https://doi.org/10.7554/eLife.33144.003

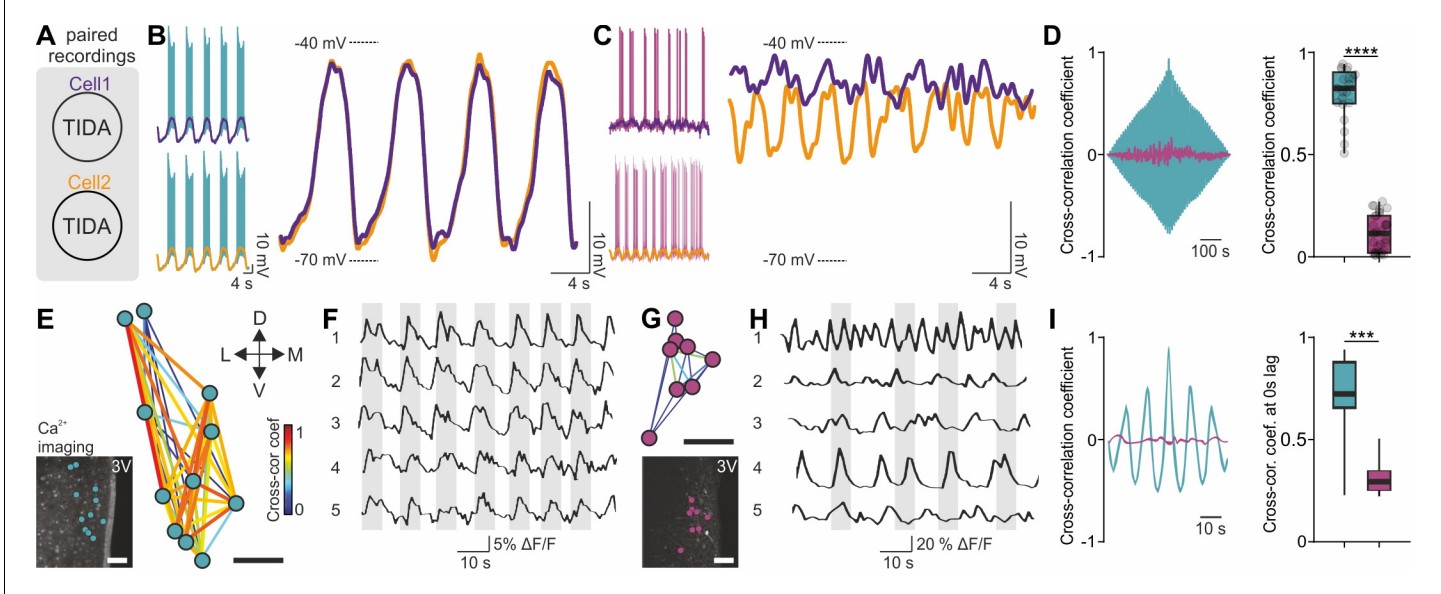

**Figure 3.** Synchronous vs asynchronous activity in TIDA neurons between two rodent species. (A) Paired whole-cell recordings were performed to test synchronous activity between cells. (B) Representative paired rat TIDA neuron recordings (left) shown superimposed and filtered for action potentials at right to reveal near identical membrane voltage fluctuations. (C) Representative paired mouse TIDA neuron recordings (left) reveal asynchronicity when superimposed (right). (D) Left; cross-correlation coefficient in example paired recordings from rat (blue) and mouse (purple) TIDA neurons. Right; quantification of cross-correlation coefficient at 0 s lag (n = 21–31 respectively, p<0.0001, unpaired t-test). (E) Ca[2+] imaging of rat TIDA neurons. Lower inset: oscillating cells in the dmArc slice shown highly connected by relative cross-correlation. (F) Representative Ca[2+] imaging traces of five rat TIDA neurons in same slice revealing synchronous oscillations. (G) Ca[2+] imaging of mouse TIDA neurons. Lower inset: Cells in the dmArc in the same slice exhibit minimal cross-correlation. (H) Representative Ca[2+] imaging traces of five mouse TIDA neurons in same slice revealing asynchronous oscillation. (I) Left; cross-correlation coefficient in rat (blue) Ca[2+] imaging traces vs cross-correlation coefficient in mouse (purple) TIDA neurons. Right; quantification of cross correlation coefficient at 0 s lag (n = 14–21 respectively). Data expressed as mean ±s.e.m. Image scale bars (E G) 50 μm. 3V, third ventricle.

DOI: https://doi.org/10.7554/eLife.33144.004

The following figure supplement is available for figure 3:

**Figure supplement 1.** Ca[2+]oscillations correlate highly with membrane voltage oscillations in TIDA neurons.
DOI: https://doi.org/10.7554/eLife.33144.005

in the TIDA circuit. Earlier work (*Lyons et al., 2010*) has suggested the existence of gap junctions in rat TIDA neurons, but direct evidence for electrical coupling has been lacking.

Recordings of sub-threshold voltage transfer (*Bennett, 1966*) in rat TIDA neuron pairs revealed electrical coupling in 51% (n = 32/63) of pairs with a high coupling coefficient (CC = 0.18 ± 0.02; *Figure 5A–F*). This electrical coupling is in the highest range reported in the mammalian brain (*Curti et al., 2012*), with single pairs reaching as high as 0.48 CC. Coupling was typically asymmetrical (strongest: weakest ratio, 1.52 ± 0.17; n = 16; *Figure 5F*, top). In notable contrast, mouse TIDA neuron pairs (n = 40) showed no evidence of electrical coupling and/or synchrony (*Figure 5G–L*). In addition, immunostaining for the dominant neuronal gap junction-forming protein connexin 36 (Cx36; *Rash et al., 2000*) was performed. Punctate Cx36 immunoreactivity was found distributed along rat (n = 4 animals), but not mouse (n = 7 animals), TH-ir dendrites and somata (*Figure 5M,N*), even though the antibody readily detects Cx36 in other mouse brain populations (*Figure 5—figure supplement 1*). These data identify an absolute species difference in TIDA gap junction connectivity. The lack of electrical coupling in the mouse TIDA neurons can also explain their higher input resistance (*Figure 4C,D,G,H*; cf. *De Zeeuw et al., 2003*), and their more fragile oscillation dynamics (*Figure 1C,F,I–T*).

These findings challenge common assumptions that analogous populations in the rat and mouse brains obey similar connectivity schemes by identifying a binary difference in electrical coupling between these two commonly studied model species. The evolutionary and physiological implications of this contrariety remain to be explored, but given that TIDA neurons control prolactin

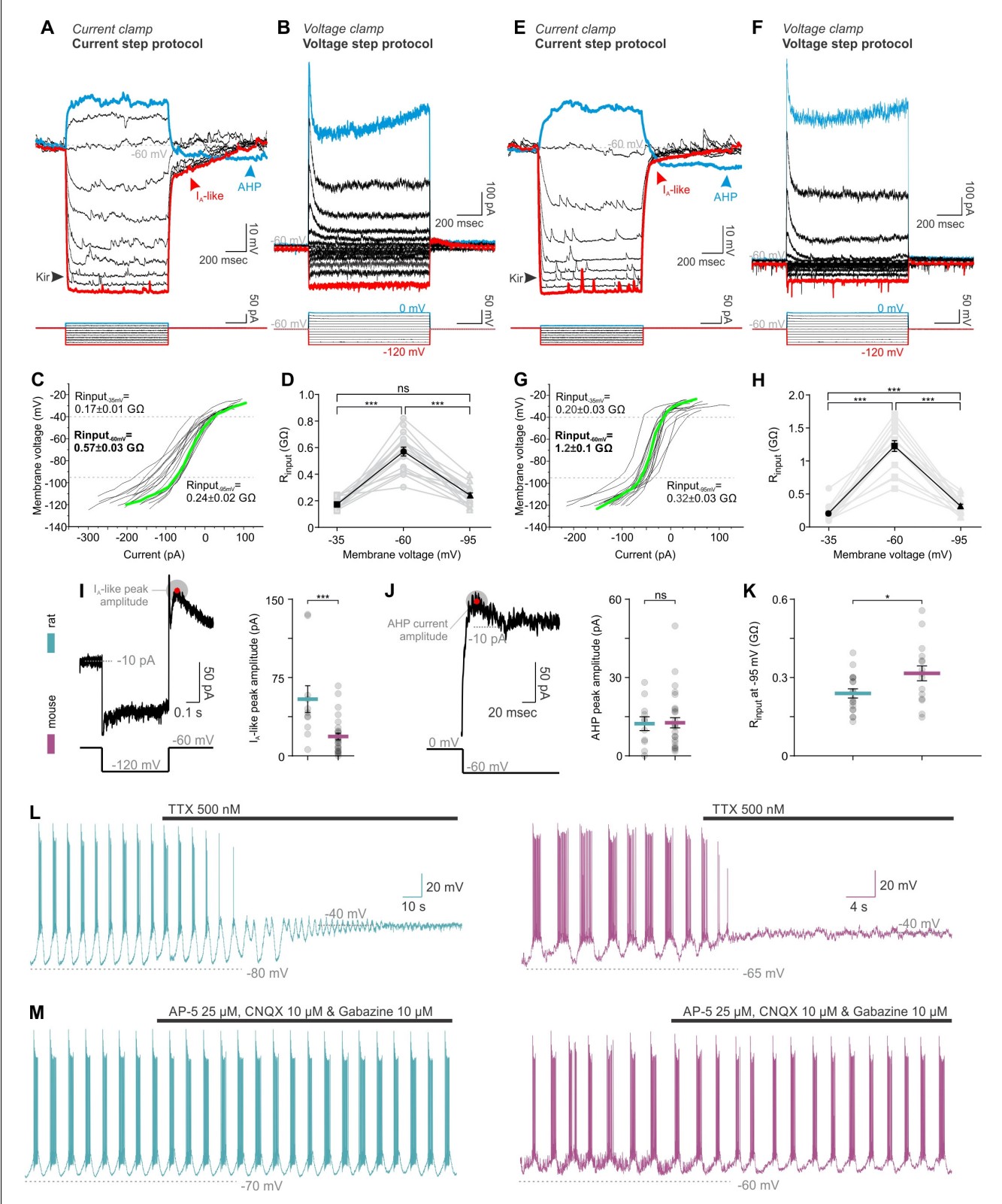

**Figure 4.** TIDA neurons have similar membrane properties, and exhibit similar sensitivity to tetrodotoxin (TTX) and synaptic blockade in both rats and mice. (A,E) Current and (B,F) voltage step protocols identify characteristic rat (A–D) and mouse (E–H) TIDA cell conductances, namely the $K^+$-mediated inward rectification ($K_{ir}$), an A-like current ($I_A$) and the depolarization-induced after hyperpolarization (AHP). (,G) Input resistance of TIDA neurons across membrane voltage reveals a sigmoidal plot in both species. (D,H) Higher input resistance at −60 mV in both rat (n = 18) and mouse (n = 16) TIDA

*Figure 4 continued on next page*

*Figure 4 continued*

neurons (one-way ANOVA with Bonferroni as post-hoc test). (I) Quantification of the A-like current and (J) the AHP current (n = 11 and 30 per group). (K) Comparison of the $R_{input}$ at −95 mV, as an indicator of the $K_{ir}$ current activity. (L) Application of the antagonist of voltage-gated $Na^+$ channels, TTX, abolished the oscillation and switched it to a persistent UP state in rat (blue; n = 32) as well as in mouse (purple; n = 22) TIDA neurons. (M) Oscillation persists in both rat (n = 11) and mouse (n = 6) TIDA neurons under fast ionotropic blockade, using AMPA and NMDA glutamate receptor blockers combined with Gabazine as a $GABA_A$ receptor blocker. (A–D), rat and (E–H), mouse.

DOI: https://doi.org/10.7554/eLife.33144.006

The following figure supplement is available for figure 4:

**Figure supplement 1.** Comparison of TIDA neuron cytoarchitecture between rat and mouse.

DOI: https://doi.org/10.7554/eLife.33144.007

secretion (*Grattan, 2015*), it is likely to have bearing on for example species diversity in reproduction and parenting (*Robitaille and Bovet, 1976*; *Bronson, 1979b*; *Bronson, 1979a*; *Limonta et al., 1985*; *Lonstein and De Vries, 2000*; *Lee et al., 2006*). At the network level, our data reconcile the discrepancies in reports of rat (*Lyons et al., 2010*; *Stagkourakis et al., 2016*) and mouse (*Brown et al., 2012*; *Romanò et al., 2013*) TIDA electrical behavior. Importantly, the rodent TIDA species difference in electrical coupling offers an experimental model to address functional aspects of the role of gap junctions in rhythmically active systems, in a manner that avoids the poor pharmacological specificity of gap junction-blocking compounds (e.g. *Beaumont and Maccaferri, 2011*) and the potential for residual electrical coupling in Cx gene-deleted mice (*De Zeeuw et al., 2003*; *Lee et al., 2014*).

We therefore next used the rat and mouse TIDA networks to address the poorly understood questions if gap junctions play a role in selecting for network frequency, and if and how neurons with heterogeneous frequencies are accommodated within a tightly electrically coupled network. For this purpose, we applied the ZAP function protocol (*Puil et al., 1986*; *Puil et al., 1994*; *Tohidi and Nadim, 2009*; *Tseng and Nadim, 2010*) to identify the preferred (or intrinsic resonance) frequency of individual cells, a feature intimately linked to oscillatory potential (*Lampl and Yarom, 1997*; *Hutcheon and Yarom, 2000a*). The ZAP protocol was applied to TIDA neurons (in the presence of TTX to remove any confounding synaptic influence; it should be noted that this pharmacological manipulation also removes the influence of the persistent $Na^+$ current on resonance) to identify their preferred frequencies, which were then compared to spontaneous oscillation frequencies recorded from the same cells (*Figure 6A*). In mouse TIDA neurons, preferred (0.35 ± 0.03 Hz; n = 15; *Figure 6E,F*) and oscillation (0.39 ± 0.05 Hz; n = 15) frequencies were not significantly different (*Figure 6G*) and were distributed over a similar spectrum (typically 0.2–0.6 Hz), suggesting that mouse TIDA cells set their own pace. Intriguingly, the preferred frequencies in rat TIDA neurons showed a similarly wide range and mean (0.30 ± 0.02 Hz; n = 32; *Figure 6B,C*) as their mouse counterparts. Yet, the rat oscillation frequencies, as described above, were clustered within a narrow interval at a significantly lower average value (0.14 ± 0.006 Hz; rat oscillation vs. preferred frequency p<0.0001; paired *t*-test; *Figure 6D*).

This observation suggests that electrical coupling places constraints on the network oscillation frequency emerging from cells with variable preferred frequencies. We therefore hypothesized that gap junction transmission, via passive or active filtering mediated through pre- and post-junctional neurites, allows current to flow at higher amplitude at certain frequencies. To test this possibility, the ZAP protocol was now applied in electrically connected pairs with high CC (>0.2) where the ZAP function was injected into one ('prejunctional') neuron and the resultant voltage response (from which resonance can be extracted; *Figure 7A,B*) was recorded from both the pre-junctional cell and its coupled ('post-junctional') partner. Recordings were performed in TTX to ensure that only gap junction transmission served as a conduit between cells. The protocol was applied in both directions, sequentially. Using this configuration, prejunctional resonance (0.31 ± 0.01 Hz; *Figure 7C*) was found to be higher than postjunctional resonance (0.17 ± 0.02 Hz, *Figure 7D*), which, in turn, was not significantly different from oscillation frequency (0.13 ± 0.01 Hz; p>0.05, *Figure 7E,F*). Notably, both postjunctional resonance and oscillation frequency were clustered within a narrow interval, in contrast to the broad range of prejunctional resonance frequencies (*Figure 7E,F*). These findings directly implicate gap junction transmission in the harmonization of faster cell frequencies to a slow population rhythm.

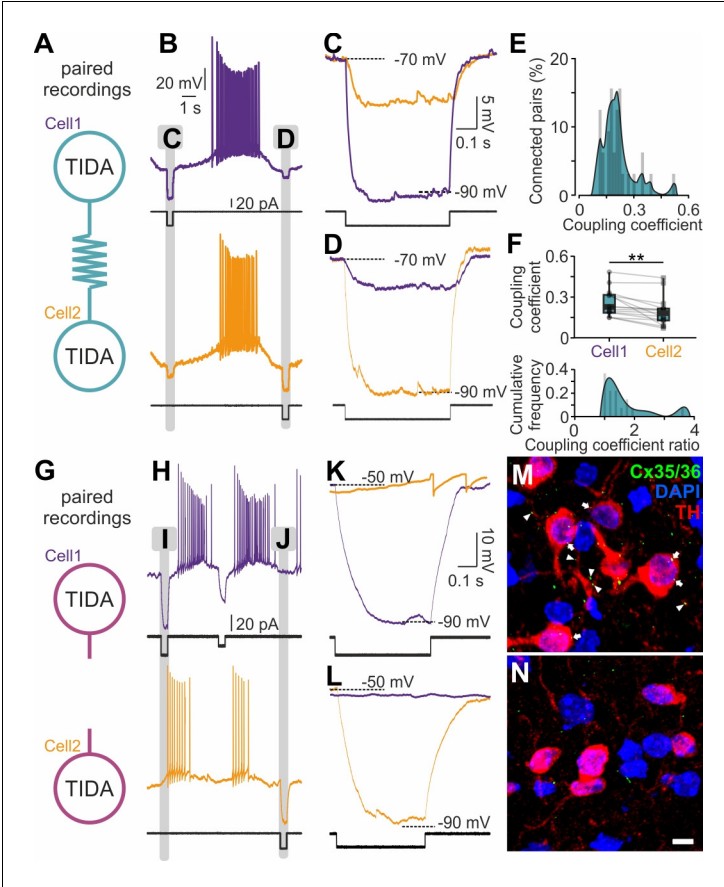

**Figure 5.** Electrical synapses in the rat but not in the mouse TIDA network. (**A**) Paired TIDA neuron recordings were performed to test for reciprocal electrical connectivity in rat slices. (**B**) Representative recording showing bidirectional electrical coupling in rat slices. Example trace illustrating the effect of negative current injection first into cell one and then into cell 2. (**C,D,**) High-resolution trace during negative current injection. Note the different response amplitude in cell1 *vs* cell2, suggestive of asymmetrical electrical synapses. (**E**) Frequency distribution of the coupling coefficient, average 0.18 ± 0.02, n = 32 pairs. (**F**) Top; Different coupling coefficient between cell 1 and cell 2, indicative of asymmetrical electrical synapses (n = 16, p<0.01, unpaired *t*-test). Bottom; Cumulative frequency of the coupling coefficient ratio, indicative of the high-degree of functional coupling asymmetry. (**G**) Similar to (**A**) paired recordings were performed in mouse TIDA neurons. (**H**) Representative recording showing the absence of electrical coupling in mouse slices. Example trace illustrating the effect of negative current injection first into cell 1 and then into cell 2. (**I** and **J**) indicate portions of recordings illustrated at higher resolution in (**K**) and (**L**), respectively. (**K, L,**) High-resolution traces illustrating lack of electrical coupling between mouse TIDA neurons (n = 40 pairs). (**M, N,**) Cx35/36 immunostaining (green) of the dmArc shows punctate presumed gap junctions (arrows) on the soma and dendrites of rat (**M**), but not mouse (**N**), TIDA neurons. TH (red) used to identify TIDA neurons, DAPI (blue) shows cell nuclei. Scale bars for (**M N**) 10 μm.
DOI: https://doi.org/10.7554/eLife.33144.008

The following figure supplement is available for figure 5:

**Figure supplement 1.** Cx35/36 immunostaining in the inferior olive and cingulate cortex in rat and mouse.
DOI: https://doi.org/10.7554/eLife.33144.009

## Discussion

In the present study, a naturally occurring absolute difference in electrical coupling among hypothalamic rodent TIDA neurons was used to investigate the role of electrical junctions in network rhythmicity. First, we show that rat (electrically coupled) and mouse (electrically uncoupled) TIDA neurons exhibit fundamentally different oscillatory activity: while electrically coupled TIDA cells discharge in a stereotyped, highly regular, synchronized slow rhythm, the same neurons in the absence of electrical coupling, display a range of faster, asynchronous, and less regular frequencies. We exploit these

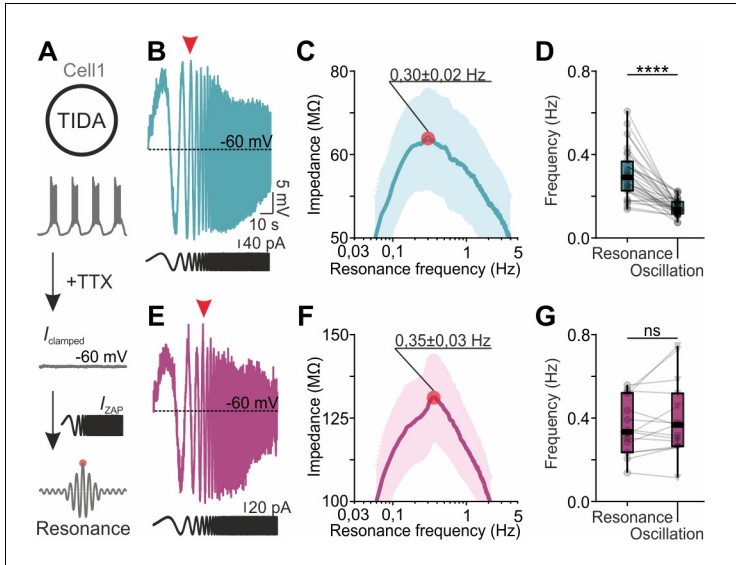

**Figure 6.** TIDA resonance frequency matches oscillation frequency in the electrically uncoupled, but not in the coupled, network. (**A**) Schematic illustration of the current clamp protocol used to determine the resonance frequency of TIDA neurons. (**B**) Voltage response (blue) of a rat TIDA neuron to a 100 s 0.05–10 Hz ZAP current injection (black trace below) yields a maximum amplitude at the neuron's preferred resonance frequency (red arrow). (**C**) Impedance profile of rat TIDA neurons in the frequency domain. The red circle indicates the resonance frequency at the maximum impedance value ($0.30 \pm 0.02$ Hz, n = 32). (**D**) Quantification of resonance frequency vs oscillation frequency in rat TIDA neurons (n = 32, p<0.0001, paired *t*-test). (**E**) Recording of mouse preferred resonance frequency as in (**B**). (**F**) Impedance profile of mouse TIDA neurons in the frequency domain. The red circle indicates the resonance frequency at the maximum impedance value ($0.35 \pm 0.03$ Hz, n = 15). (**G**) Resonance frequency vs oscillation frequency in mouse TIDA neurons (n = 15, p>0.05, paired *t*-test).
DOI: https://doi.org/10.7554/eLife.33144.010

differences to show that, while both species harbor a similar range of cellular heterogeneity within the TIDA population (assessed as preferred frequency and membrane properties), the presence of gap junctions imposes a slow, harmonized, narrow frequency band across the ensemble. The main findings and conclusions are summarized in *Figure 8*.

Computational studies have proposed a variety of rules for how frequency is set in an oscillatory electrically coupled network (*Kepler et al., 1990*; *Meunier, 1992*; *Pedersen et al., 2005*). Our experimental data show that in a gap junction-linked hypothalamic dopamine circuit, the network assumes a harmonized slow rhythm emerging from the preferred resonance frequency imposed by the electrical coupling. Early studies of neuronal gap junctions suggested that they operate as low-pass current filters (*Furshpan and Potter, 1957*). As oscillatory current passes between cells through gap junctions – and filtered via the active and passive properties of membranes on either side of the junction – the variable frequencies generated by individual TIDA neurons are restricted and streamlined toward a slow network frequency. The mechanistic demonstration of the role of gap junctions in current transfer (*Figure 7*) argues against the interpretation that the low frequency of the rhythm in this system is the result of 'network load' (*Hooper and Marder, 1987*; *Meunier, 1992*), the drag imposed upon a connected neural ensemble by its slowest members, whose biophysical constraints may act as a brake on the entire network. The specific frequency emerging out of the electrically coupled network is likely not to be an exclusive product of the biophysical properties of the proteins (in this case Cx36) that form the conducting connexons; there have been several Cx36-linked rhythmically active populations described in the CNS, which oscillate in a wide range of frequencies. Other factors specific to the coupled population, for example the subcellular distribution of gap junctions, cellular geometry and associated proteins, may be hypothesized to contribute.

These findings also allow for some observations on biological diversity, by revealing how two related species have adopted different cellular solutions to a common biological problem – the adaptive control of pituitary prolactin secretion. While the two rodent TIDA populations responsible

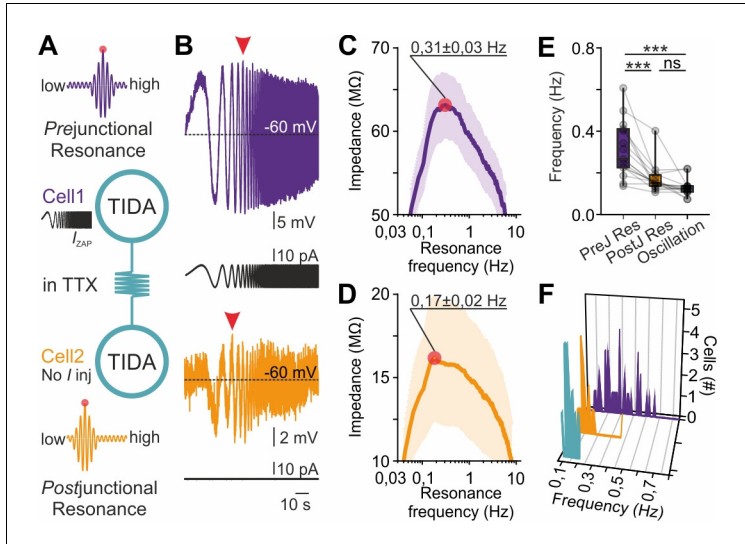

**Figure 7.** Gap junction resonance frequency dictates network frequency of coupled heterogeneous TIDA neurons. (A) Schematic of the current clamp protocol used to record the transfer of a ZAP function command through electrical synapses by measurements of resonance frequency in coupled TIDA neurons. (B) Paired recording of electrically connected rat TIDA neurons. Voltage response of Cell 1 (purple; prejunctional) to the 100 s ZAP current injection that sweeps a 0.05–10 Hz frequency (black trace) range, yields a maximum amplitude at a preferred resonance frequency. Cell 2 membrane voltage response (orange; postjunctional) indicates maximum amplitude at a different resonance frequency. (C) The impedance profile of the prejunctional resonance in the frequency domain. The red circle indicates the resonance frequency at the maximum impedance value (0.31 ± 0.03 Hz; n = 16). (D) Postjunctional resonance impedance profile; note lower average resonance frequency (red circle; 0.17 ± 0.02 Hz; n = 16). (E) Comparison of pre- and post-junctional resonance frequency vs oscillation frequency (n = 16, one-way ANOVA with Bonferroni as post-hoc test). (F) Frequency distribution of pre- (purple) and post-junctional (orange) resonance frequency vs oscillation frequency (turquoise) (n = 16 per group).

DOI: https://doi.org/10.7554/eLife.33144.011

for this control accommodate a similar range of cellular heterogeneity among their intrinsic oscillators, the connectivity that imposes network rules thus differs critically (*Figure 8*). In the rat TIDA network, cellular diversity is shown to be accommodated through a powerful gap junction connectivity. Cellular diversity also within seemingly homogeneous neuronal populations has been demonstrated to be not only a core characteristic of how nervous systems are organized, but a feature that contributes critically to the flexibility and adaptability necessary for survival under variable environmental conditions (*Manor et al., 1997a*; *Golowasch et al., 1999*; *Manor et al., 2000*; *Prinz et al., 2004*; *Grashow et al., 2010*; see *Marder et al., 2015*). How the different configurations (electrically connected vs. not connected) impact on the physiology of the two species remains to be determined, but it may be speculated that these features are of particular potential benefit in a system tasked with adjusting the animal for recurrent reproductive stages, including pregnancy, and is subject to circadian modulation, as is the case for the TIDA-prolactin axis (*Grattan et al., 2008*).

Diversity within neuronal networks, as described here in the TIDA system, presents a trade-off: while it increases flexibility, it also risks compromising reliable circuit performance. One means of ensuring robustness is thus by adding constraining properties in the mode(s) whereby the cells are connected. We here provide evidence that connecting intrinsically oscillating neurons together via strong electrical coupling can impose a slow network rhythm that overrides the individual frequencies of cells within the population. This work suggests a principle whose applicability to other systems can now be addressed.

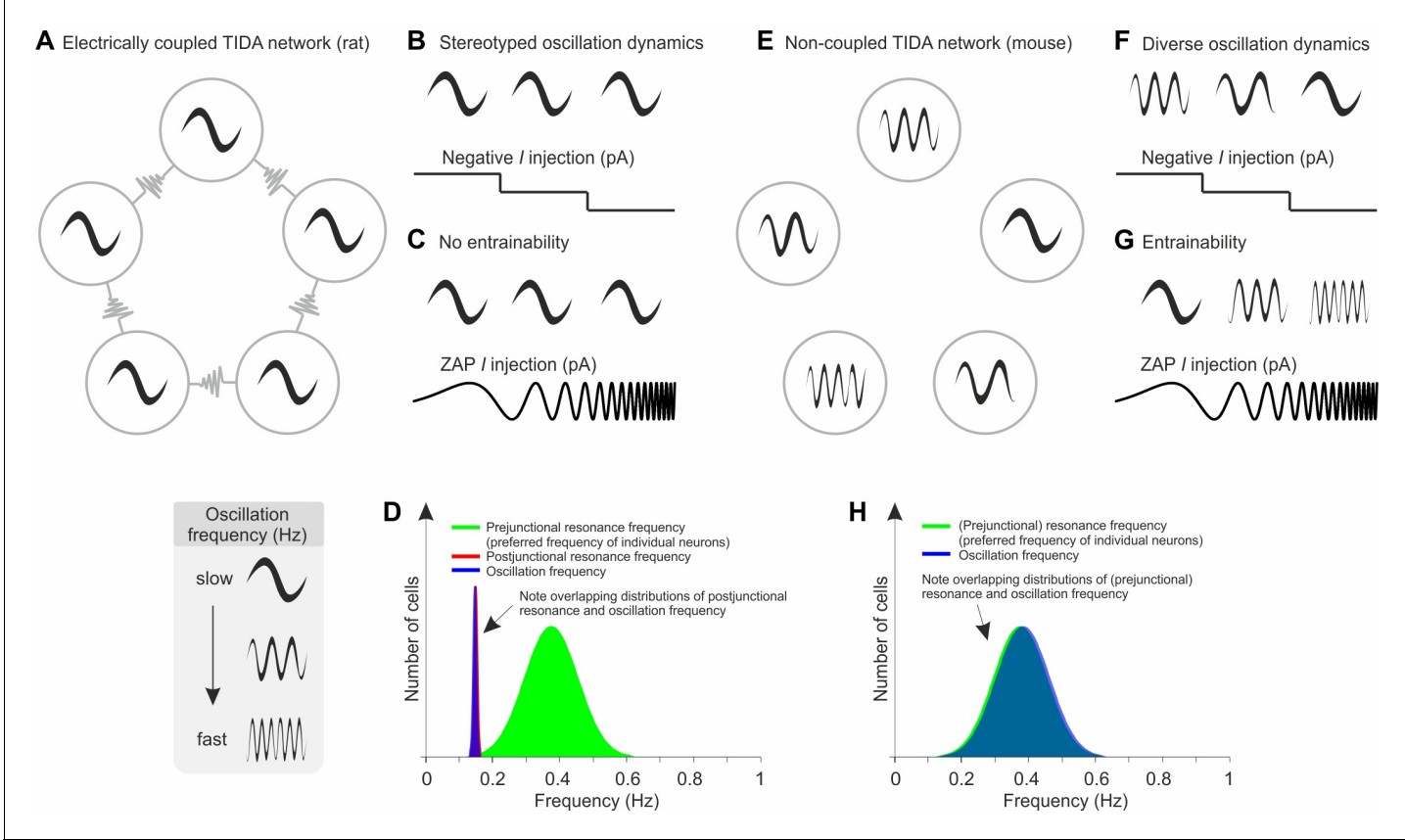

**Figure 8.** Schematic model based on the experimental data from TIDA neurons of the influence of electrical synapses on resonance and the emergence of a slow frequency oscillation in a neuronal population. (**A**) In the electrically coupled network (as observed in rat TIDA neurons), neurons exhibit a robust, stereotyped, slow, synchronized oscillation. In this network, frequency is immune to perturbation and cannot be manipulated by progressive hyperpolarization (**B**) or by ZAP current injection (**C**). (**D**) The prejunctional resonance of individual neurons does not dictate the network oscillation frequency; rather, the gap junction-mediated postjunctional resonance commands the oscillation frequency of the syncytium. (**E**) In the uncoupled network configuration (as observed in mouse TIDA neurons), neurons exhibit a diverse range of membrane potential oscillations. Their phasic voltage fluctuations are strongly sensitive to progressive hyperpolarization (**F**) and are highly entrainable, as tested via ZAP current injections (**G**). (**H**) The fast prejunctional resonance of individual neurons matches their oscillation frequency, allowing them to march to the beat of their own intrinsic properties.

DOI: https://doi.org/10.7554/eLife.33144.012

# Materials and methods

## Key resources table

| Reagent type (species) or resource | Designation | Source of reference | Identifiers | Additional information |
|---|---|---|---|---|
| Mouse strain, strain background | DAT-Cre mouse (C57Bl/6J) | Reference: PMID 17227870. | | Gift from Dr. N-G Larsson, Karolinska Institutet. |
| Mouse strain, strain background | tdTomato floxed mouse (C57Bl/6J) | www.jax.org/strain/007905 | | The Jackson Laboratory |
| Mouse strain, strain background | GCaMP3 floxed mouse (C57Bl/6J) | www.jax.org/strain/014538 | | The Jackson Laboratory |
| Antibody | anti-tyrosine hydroxylase | Source: Millipore. Reference: PMID 25794171 | AB152 | |
| Antibody | anti-connexin 35/36 | Source: Millipore. Reference: PMID 26760208. | MAB3045 | |
| Software, algorithm | MATLAB code for network analysis | Source: PMID 25278844 | | |

Key reagents and their sources are listed in the Key Resources Table. The data that support the findings of this study are available from the corresponding author upon reasonable request.

## Animals

All animal experiments were performed on male animals on postnatal days 21–28, had received approval by the local ethical board, *Stockholms Norra Djurförsöksetiska Nämnd*, and were performed in accordance with the European Communities Council Directive of November 24, 1986 (86/609/EEC). Sprague Dawley rats (Charles River), DAT-Cre-tdTomato-floxed and DAT-Cre-GCaMP3-floxed C57Bl/6J mice (own breeding – described in [*Ekstrand et al., 2007*]) were housed with *ad libitum* access to standard chow and tap water in a temperature-controlled environment under 12 hr light/dark conditions with lights on at 6:00 A.M.

## Whole-cell recordings

For electrophysiological experiments, animals were deeply anesthetized with sodium pentobarbital and decapitated. Only one experiment was performed on each slice. The brain was rapidly removed and placed in ice-cold (2–4°C) and oxygenated (95%$O_2$/5%$CO_2$) slicing solution containing the following (in mM): 214 sucrose, 26 $NaHCO_3$, 10 D-glucose, 1.2 $NaH_2PO_4$, 2.0 KCl, 1.3 $MgSO_4$ and 2.4 $CaCl_2$. A modification of the technique developed by *Aghajanian and Rasmussen (1989)* was used to increase tissue viability. The meninges were gently removed, and the brain was blocked and glued to a vibratome (Leica VT-1000) in which 250 µm-thick coronal sections of the hypothalamus containing the dmArc were prepared unless otherwise stated. After slicing, tissue was immediately transferred to an extracellular recording solution (aCSF) containing (in mM): 127 NaCl, 26 $NaHCO_3$, 10 D-glucose, 1.2 $NaH_2PO_4$, 2.0 KCl, 1.3 $MgSO_4$, 2.4 $CaCl_2$, in a continuously oxygenated holding chamber at 35°C for a period of 30 min. A minimum of 1 hr of recovery was allowed after which intracellular whole-cell single unit recordings were performed from TIDA neurons located in the dmArc. For whole-cell recordings, slices were transferred to a submerged chamber and placed on an elevated grid that allows perfusion both above and below the slice. An Axio Examiner D1 upright microscope (Carl Zeiss) was used for infrared differential interference contrast visualization of cells. Recordings were performed at near-physiological temperature (34 ± 1°C) apart from subset of experiments described in *Figure 2* performed in room temperature (24 ± 1°C), and slices were continuously perfused with oxygenated recording solution at a rate of 4 ml/min. Whole-cell current- and voltage-clamp recordings were performed with pipettes (3–7 MΩ when filled with intracellular solution) made from borosilicate glass capillaries (World Precision Instruments) pulled on a P-97 Flaming/Brown micropipette puller (Sutter Instruments). The intracellular recording solution used in experiments contained (in mM) 140 K-gluconate, 10 KCl, 10 HEPES, 1 EGTA, and 2 $Na_2ATP$, pH 7.3 (with KOH). All pharmacological compounds were bath applied. Blockade of ionotropic glutamatergic and GABAergic transmission was achieved by adding 10 µM of the AMPA/kainic acid antagonist 6-cyano-7-nitroquinoxaline-2,3-dione (CNQX), 25 µM of the NMDA antagonist DL-2-amino-5-phosphonopentanoic acid (AP5) and 10 µM of gabazine (all from Abcam) to the extracellular recording solution. The concentration used for TTX (Alomone Labs) was 0.5 µM. (-)-quinpirole hydrochloride, 5-HT, prolactin and TRH were obtained from Sigma.

## Paired recordings and electrical coupling

The strength of coupling between neurons was quantified by the coupling coefficient (CC), which was typically measured by injecting a 0.5 s current step to one neuron and measuring the resulting voltage deflections in both neurons. To minimize distortion of CC by voltage-activated conductances, the current step amplitude was set sufficiently large to produce a steady voltage response in the coupled neuron. For TIDA neurons with input resistance ($R_{input}$) of 500–700 MΩ and a minimum CC of 0.05, we used 20–40 pA current injections while both cells were entering the DOWN state (phase1) resulting in 20–40 mV presynaptic deflections (ΔV) in the injected cell, resulting in stable deflections (ΔV) equal to or larger than 1 mV in the postjunctional cell. 50-sweep negative pulse protocols were routinely used allowing quantification of CC at different voltages via sweep averages. CC was calculated at different voltages, as conductances can amplify/dampen both prejunctional and postjunctional measurements and coupling (*Curti et al., 2012*; *Haas and Landisman, 2011*). For

current injection $I_1$ into cell 1, the coupling measured in cell 2 is $CC_{12}=\Delta V_2/\Delta V_1$ and vice versa for $CC_{21}$. The quantity $CC_{12}$ measures coupling in the direction of cell 2. Postjunctional resistance between pairs was $80 \pm 10$ M$\Omega$ (n = 10).

*Recordings* were performed using a Multiclamp 700B amplifier, a DigiData 1440 and pClamp10.2 software (Molecular Devices). Slow and fast capacitive components were automatically compensated for. Access resistance was monitored throughout the experiments, and neurons in which the series resistance was more than 25 M$\Omega$ or changed 15% were excluded from the statistics. Liquid junction potential was 16.4 mV and not compensated. Sampling frequency in all electrophysiology data was 10 kHz. Raw data in figures were filtered at 1 kHz or 2 kHz for illustrative purposes.

## Logarithmic ZAP function

To investigate the possibility of resonance/preferred frequency of TIDA neurons, we performed sinusoidal current injections of fixed amplitude but sweeping frequencies in a given range. The impedance (Z) Amplitude Profile (ZAP) was thus generated for TIDA neurons to determine neurons' preferred frequency (*Tseng and Nadim, 2010*) generated by a combination of passive and active properties (*Puil et al., 1986*). The ZAP current is described as:

$$I_{ZAP} = I_{max}\sin(2\pi f(t)x\,t)$$

where *f(t)* produces the range of frequencies that the ZAP function sweeps. We used an exponential chirp function for *f(t)* to increase the sampling duration of lower frequencies:

$$f(t) = \frac{f_{min}}{L}\left(e^{Lt} - 1\right)$$

where

$$L[log(fmax/fmin)] \, / \, total\ chirp\ duration$$

As an optimal frequency range of the ZAP function input allowing accurate determination of the TIDA impedance characteristics, $f_{min}$ was set to 0.05 Hz and $f_{max}$ as 10 Hz. The amplitude of the ZAP current was selected as the minimum amplitude that produced a voltage response within the physiological range of the TIDA neuron subthreshold oscillation ($-80$ mV to $-40$ mV). This amplitude was typically set to 15–20 pA. Extra caution was placed at targeting resonance within close limits of the membrane voltage, since resonance is described to be voltage dependent (*Puil et al., 1987*). This protocol was performed in the presence of TTX, resulting in a resting membrane potential and eliminating AP interference with likely resonance mechanisms. A bias direct current (DC) of ~$-20$ pA was added to compensate for the depolarizing action of TTX. Furthermore, to avoid transients at the beginning of the oscillations, the ZAP waveform was preceded with three cycles of a sinusoidal waveform of the $f_{min}$ before transition into the ZAP function.

The impedance of neurons was calculated as a function of the frequency of the injected ZAP current. Impedance contains information about both amplitude and phase of oscillation and is calculated as a function of the input frequency as follows:

$$Z(f) = \frac{\tilde{V}(f)}{\tilde{I}(f)}$$

where $\tilde{V}$ and $\tilde{I}$ are, the Fourier transforms of the membrane potential *V* and the ZAP current *I*, respectively (Hutcheon and Yarom, 2000b). *Z(f)* is a complex number; the absolute value of *Z* is the impedance power and is plotted as a function of frequency. For simplicity here we ignore phase, and in our results, we refer to the impedance power as impedance. Lastly, as negative control for the resonance experiments we analyzed the impedance of a passive neuron showing, as expected, no peak. Additionally, as previous studies suggest (Puil et al., 1994), we tested and concluded that direction (slow-to-fast vs fast-to-slow frequencies) and frequency spectrum (0.05 - 3 Hz *vs* 0.05 – 10 Hz) of the ZAP current do not alter the TIDA neuron resonance frequency.

### Entrainment of TIDA neurons

To test entrainability of TIDA neurons, the ZAP function was applied in the 0.05–3 Hz frequency spectrum. The amplitude of the current was determined individually per neuron, as the amount of negative current required to induce a $-10$ mV $\Delta$V in the DOWN state (phase 1).

### Dye loading and Ca$^{2+}$ imaging

For bulk loading of TIDA neurons in rats, the Ca$^{2+}$-sensitive dye Oregon Green BAPTA-1 AM (OGB-1 AM, Invitrogen) was first dissolved in 0.5% DMSO, 0.01% Pluronic F-127 (Molecular Probes), 0.002% Cremophor (Sigma-Aldrich) and further diluted in aCSF to a final concentration of 20 μM. Slices were incubated for 40 min in oxygenated aCSF containing the cell-permeable OGB-1 AM dye solution. The activity of oscillating neurons of rats (AM dye) and mice (DAT-GCaMP3) was monitored by imaging fluorescence changes under visual control by two-photon imaging and CCD camera (water immersion objective, Zeiss) at ~900 nm. Scanning and image acquisition were controlled by custom software (LSM) during periods of 120 s at intervals of 300 ms. Imaging frames were acquired at 3 Hz. Image sequences were analyzed with custom programs written in ImageJ (NIH) and MATLAB (*Smedler et al., 2014*). The recorded intensity signals are presented as relative changes of fluorescence in each of the selected ROIs and expressed as ($\Delta F/F_0$), where $F_0$ is the baseline fluorescence measured over a period of low activity of each cell. Simultaneous Ca$^{2+}$ imaging and whole-cell recordings (*Figure 3—figure supplement 1*) were performed on a Zeiss microscope system (Carl Zeiss, Germany). Fluorescence images were captured with a CCD camera (Evolve, Photometrics) during periods of 120 s at intervals of 300 ms. A 64x water-dipping objective was used when a neuron was being simultaneously patched. In these cases, no EGTA was used in the intracellular solution. Whole-cell recordings were performed through visualizing positive cells and then adjusting to DIC images. For temporal synchronization of acquired images and recorded membrane potential oscillations, marker pulses indicating the occurrence of each frame were obtained from the imaging system and recorded together with the intracellular signal using pClamp10.2 software.

### Immunofluorescence

Rats and mice were perfused first with pre-fixative (50 mm sodium phosphate buffer, 0.1% sodium nitrite, 0.9% NaCl and 1 unit/mL of heparin, pH 7.4) and next with cold fixative (0.16 m sodium phosphate buffer, 0.2% picric acid, 1–2% formaldehyde prepared from freshly depolymerized paraformaldehyde, pH 7.4), and finally with 10% sucrose/25 mM sodium phosphate buffer (pH 7.4), to wash out excess fixative. Tissues were stored at 4°C for 24–48 hr in cryoprotectant (25 mM sodium phosphate buffer, 10% sucrose, 0.04% sodium azide, pH 7.4). Tissue sections (14 μm) were processed for conventional immunofluorescence, using primary anti-connexin-35/36 (Cx36) immunoglobulin (1:2000; mouse monoclonal, MAB3045, Millipore), combined with anti-tyrosine hydroxylase (TH) antiserum (1:2000; raised in rabbit, AB152, Millipore), counterstained with DAPI and mounted with ProLong Gold (ThermoFisher). Confocal micrographs were acquired with an Olympus FV1000 microscope and analyzed and processed in BitPlane Imaris. For final images, brightness, contrast and sharpness were adjusted digitally.

### Cell filling and reconstruction

Mouse (n = 5 cells from 4 animals) and rat (n = 5 cells from 5 animals) TIDA neurons were recorded in whole-cell mode with intracellular pipette solution as above, with the addition of 0.2% neurobiotin. After recording, slices were placed in fixative (4% paraformaldehyde/0.16% picric acid), washed in PBS and incubated at 4°C for 72 hr in a solution containing FITC-conjugated avidin (1:2500, 43–4411, Zymed) and mouse anti-tyrosine hydroxylase immunoglobulin (1:2000, MAB318, Millipore). After extensive washing, slices were incubated in secondary Alexa-594-conjugated donkey-anti-mouse antiserum (1:500; Invitrogen, Carlsbad, CA; A21202) for 16 hr at 4°C, washed again and mounted with 2.5% DABCO in glycerol. TIDA identity of all filled cells was confirmed with TH immunoreactivity. Cells were digitally reconstructed in MBF Neurolucida using a Zeiss Axio Imager M1 with a 63x objective and morphological parameters were extracted from the virtual neurons with MBF Neurolucida Explorer. Sholl analysis was performed by counting the number of line crossings on concentric circles spaced 20 μm apart.

## Data analysis

Electrophysiological data analysis was performed with Clampfit (Molecular Devices), OriginPro8 (OriginLab) and custom written MatLab routines. Statistical analysis was performed using GraphPad Prism6, and statistical significance was set at $p<0.05$. All data are presented as means ± SEM. Significance levels used in figures are shown as: single asterisk (*)=$p < 0.05$, double asterisks (**)=$p < 0.01$ and triple asterisks (***) when $p<0.001$. Results were analyzed using the unpaired two-tailed Student's t-test unless otherwise stated. In sections stating that data were analyzed using one-way ANOVA, post-hoc analysis was performed with Tukey's range test.

## Acknowledgements

We thank Drs. J Ausborn, H Kim and V Caggiano, and J van Lunteren and P Williams for assistance and advice with data analysis. Dr. I Dehnisch is acknowledged for generously sharing his expertise in two-photon imaging, Dr. M Gómez Galán for advice on Ca2+ indicator dyes, Dr. J I Nagy for advice on immunostaining, and Dr. N-G Larsson for sharing DAT-Cre mice.

## Additional information

### Funding

| Funder | Grant reference number | Author |
| --- | --- | --- |
| European Research Council | 261286 | Christian Broberger |
| Vetenskapsrådet | 2014-3906 | Christian Broberger |
| Karolinska Institutet | SRP Diabetes, Rolf Luft Award | Christian Broberger |
| Novo Nordisk | Project Grant | Christian Broberger |
| Hjärnfonden | Postdoctoral fellowship | Carolina Thörn Pérez |

The funders had no role in study design, data collection and interpretation, or the decision to submit the work for publication.

### Author contributions

Stefanos Stagkourakis, Carolina Thörn Pérez, Conceptualization, Formal analysis, Investigation, Visualization, Methodology, Writing—original draft, Writing—review and editing; Arash Hellysaz, Investigation, Visualization, Writing—review and editing, Formal analysis,; Rachida Ammari, Investigation, Writing—review and editing; Christian Broberger, Conceptualization, Supervision, Funding acquisition, Writing—original draft, Project administration, Writing—review and editing

### Author ORCIDs

Stefanos Stagkourakis (iD) http://orcid.org/0000-0003-1218-791X
Carolina Thörn Pérez (iD) http://orcid.org/0000-0002-3480-8599
Arash Hellysaz (iD) http://orcid.org/0000-0003-4512-3795
Christian Broberger (iD) http://orcid.org/0000-0002-7050-8809

### Ethics

Animal experimentation: All animal experiments had received approval by the local ethical board, Stockholm's Norra Djurförsöksetiska Nämnd, and were performed in accordance with the European Communities Council Directive of November 24, 1986 (86/609/EEC). All euthanasia was performed under sodium pentobarbital anesthesia, and every effort was made to minimize suffering.

### Decision letter and Author response

Decision letter https://doi.org/10.7554/eLife.33144.017
Author response https://doi.org/10.7554/eLife.33144.018

## Additional files

### Supplementary files

• Source data 1. Oscillation dynamics in rat TIDA neurons. Injection of negative current in current clamp results in hyperpolarization of the subthreshold oscillation. Note that the oscillation frequency remains unchanged irrespective of membrane voltage.

DOI: https://doi.org/10.7554/eLife.33144.013

• Source data 2. Oscillation dynamics in mouse TIDA neurons:Injection of negative current in current clamp, results in a switch from continuous firing to oscillatory discharge. Note that upon further negative current injection, the oscillation collapses at −80 mV.

DOI: https://doi.org/10.7554/eLife.33144.014

• Transparent reporting form

DOI: https://doi.org/10.7554/eLife.33144.015

### Data availability

A sample of the underlying electrophysiological data is available as Source Data Files 1 and 2. Due to the large size of the the whole dataset, it will be made available to the scientific community by the corresponding author upon reasonable request.

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
