## [Decision Letter]

Thank you for submitting your article "Network Oscillation Rules Imposed by Species-Specific Electrical Coupling" for consideration by *eLife*. Your article has been reviewed by three peer reviewers, one of whom Ronald Calabrese is a member of our Board of Reviewing Editors and the evaluation has been overseen by Eve Marder as the Senior Editor. The following individuals involved in review of your submission have agreed to reveal their identity: Farzan Nadim (Reviewer #2); Yosef Yarom (Reviewer #3).

The reviewers have discussed the reviews with one another and the Reviewing Editor has drafted this decision to help you prepare a revised submission.

Summary:

This very interesting paper reports on a species difference in electrical coupling between neuroendocrine tuberoinfundibular dopamine (TIDA) neurons in rats vs. mice. It leverages this rather simple observation into a done-by-nature experiment to show how electrical coupling can set network period in a network that shows rhythmic activity. The authors make the conclusion that the coupling in the rat system not only harmonizes network period but also acts as a filter to set a network period that is below the average period of the resonance frequency in the individual neurons. The authors use sophisticated electrophysiological methods to approach these questions and provide strong quantitative data.

The general appeal of this work derives from the important species difference uncovered, albeit the functional implications for the TIDA network are unexplored, and the very nice exploration of how network period is set in the rat TIDA network.

Essential revisions:

There are concerns that although the observations are sound that they are in some cases over interpreted.

1) Whereas the presence and absence of gap junctions is rather convincing, the conclusion that it is the sole difference between the species is not fully supported. Figure 4 is supposed to show similar membrane properties, but other than the input resistance there is no quantitative comparison and some seeming inconsistencies. For example, the current clamp records show there is little effect IA like current in mouse neurons (Figure 4E) and the voltage clamp records do not show any AHP current in mouse neurons (Figure 4F). Finally, one would like to know, if there is a difference in size of neurons between species that could account for the different in input resistance. The cells presented in Figure 5 (M and N) seems to support such a possibility. The authors should quantify as much as possible and should temper their claims about equivalence of membrane properties.

2) The use of ZAP current injection to measure the resonance properties of the neurons, demonstrating that although both types of neurons display similar resonance, the frequency of oscillation in the coupled network is lower than the resonance frequency is convincing. This was followed by an elegant experiment where cross gap-junction resonance was measured. However, the conclusion derived from this experiment is not fully justified. We do not agree that the data: "challenge the idea of electrical synapses operating merely as low pass current filters" Nor do we agree that the junction "acts as a band-pass filter for selecting and setting the oscillation frequency". In fact, the reduction in the resonance frequency across the junction, seems to conform to the low pass filtering expected due to passive or active filtering by the pre/post-junctional neurites.

[Editors' note: further revisions were requested prior to acceptance, as described below.]

Thank you for resubmitting your work entitled "Network Oscillation Rules Imposed by Species-Specific Electrical Coupling" for further consideration at *eLife*. Your revised article has been favorably evaluated by Eve Marder (Senior editor), a Reviewing editor, and two reviewers.

The manuscript has been improved but there are some remaining issues that need to be addressed before acceptance, as outlined below.

Reviewer #2:

The revisions have made the manuscript better and the overall quality of the data is commendable. I have no major comments other than my previous point of whether there is any broader message other than rats are not mice.

Reviewer #3:

The authors adequately responded to all my comments, adding new information and rephrase some of the statements. My only concern is the finding that the two types of cells differ in their IA current (by a factor of 2-3; Figure 4I). To the best of my understanding this should shift the resonance frequency of the cells to higher value and yet Figure 6 shows similar resonance. An explanation is needed here.

---

## [Author Response]

Essential revisions:There are concerns that although the observations are sound that they are in some cases over interpreted.1) Whereas the presence and absence of gap junctions is rather convincing, the conclusion that it is the sole difference between the species is not fully supported. Figure 4 is supposed to show similar membrane properties, but other than the input resistance there is no quantitative comparison and some seeming inconsistencies. For example, the current clamp records show there is little effect IA like current in mouse neurons (Figure 4E) and the voltage clamp records do not show any AHP current in mouse neurons (Figure 4F).

The referees raise the important point that singling out the binary difference in electrical coupling as the sole distinction between rat and mouse TIDA neurons may be an over-simplification. We fully agree and apologize if this was the impression given in the manuscript. In accordance with the request from the referees, we have now performed quantification of the three conductances mentioned: the A-like current, the AHP current and the I_Kir_. The numbers are now included in the updated version of Figure 4 (Figure 4I-K) and described in the Results section. This analysis revealed that the AHP current is not statistically different between the species, but that the A-like current is bigger and the I_Kir_ smaller in rat TIDA neurons. While we do not think that these data alter the main conclusions of the study (as the resonance frequency of rat and mouse TIDA neurons is similar), they enrich our understanding of how the TIDA system is different (and similar) across species, and we thank the referees for this request.

Finally, one would like to know, if there is a difference in size of neurons between species that could account for the different in input resistance. The cells presented in Figure 5 (M and N) seems to support such a possibility. The authors should quantify as much as possible and should temper their claims about equivalence of membrane properties.

As the referees rightly point out, the species difference in input resistance could derive from additional sources than the difference in electrical coupling, including differences in rat and mouse cell size. To get an estimate of how TIDA neurons compare between the species, we filled neurons during recording with neurobiotin, stained with Streptavidin and reconstructed them using Neurolucida. We measured soma surface, but also dendrite length and included Sholl analysis to get a measure of the complexity of dendritic branching; they have been included in the new Figure 4—figure supplement 1 and are described in Results section. Though these indices do not give a complete estimate of cell volume, they address several aspects of morphometry and neither of the numbers showed significant differences between the species. Thus, this is unlikely to be a major factor underlying the different input resistance numbers for the species. We agree with the referees that Figure 5M-N gave the impression (not supported by the analysis that has now been added; *vide supra*) that rat TIDA neurons may be larger than their mouse counterparts. For this purpose, we have replaced Figure 5M with a more representative example of Cx35/36-stained rat TIDA neurons.

2) The use of ZAP current injection to measure the resonance properties of the neurons, demonstrating that although both types of neurons display similar resonance, the frequency of oscillation in the coupled network is lower than the resonance frequency is convincing. This was followed by an elegant experiment where cross gap-junction resonance was measured. However, the conclusion derived from this experiment is not fully justified. We do not agree that the data: "challenge the idea of electrical synapses operating merely as low pass current filters" Nor do we agree that the junction "acts as a band-pass filter for selecting and setting the oscillation frequency". In fact, the reduction in the resonance frequency across the junction, seems to conform to the low pass filtering expected due to passive or active filtering by the pre/post-junctional neurites.

The referees’ comment prompted us to think twice about the conclusions, and we agree that the properties revealed in the gap-junction linked rat TIDA network can best be described as a low-pass filter. In our previous reasoning, we focused on the fact that rat neurons all oscillate within a very narrow interval, which individual cells essentially neither exceed nor subceed (for lack of a better term). But given the data distribution, this could still reflect a low-pass filter condition, and the data do not then negate that possibility. We have rephrased in the Abstract, Results section and the Discussion section to better reflect that our data are not disharmonious with earlier gap junction literature in this regard.

[Editors' note: further revisions were requested prior to acceptance, as described below.]

The manuscript has been improved but there are some remaining issues that need to be addressed before acceptance, as outlined below.Reviewer #2:The revisions have made the manuscript better and the overall quality of the data is commendable. I have no major comments other than my previous point of whether there is any broader message other than rats are not mice.

We are happy that the referee finds the revised version an improvement and appreciate the kind words regarding the data quality. We believe the study expands beyond the original finding of the species difference between rats and mice, and that the demonstration of how electrical coupling impacts on network frequency and the processing of cellular heterogeneity into a coherent ensemble output will be of interest to a broad audience.

Reviewer #3:The authors adequately responded to all my comments, adding new information and rephrase some of the statements. My only concern is the finding that the two types of cells differ in their IA current (by a factor of 2-3; Figure 4I). To the best of my understanding this should shift the resonance frequency of the cells to higher value and yet Figure 6 shows similar resonance. An explanation is needed here.

We agree with this point; the referee predicts that a difference in A-current amplitude should impact on the average resonance and is therefore concerned that the latter parameter is the same between rat and mouse TIDA neurons whereas the former is not. The impact of the A-current on resonance is, as far as we have been able to determine, relatively poorly explored in the literature, compared to other currents. It is therefore difficult to estimate – and discuss within the context of the present data – exactly how changes in this membrane property would affect intrinsic resonance, although a shift to higher frequencies, as suggested by the referee, is a reasonable assumption. Importantly, however, resonance the product of several membrane properties interacting and a difference in a single one of these may be counteracted by other, as yet undetermined changes.

It’s important in this context to bear in mind that rat and mouse TIDA neurons differ in input resistance. Interestingly, the rat neurons with the lower input resistance (Rinput = 0.6 GΩ) have higher IA amplitude while the mouse TIDA neurons with the higher input resistance (1.2 GΩ) have a lower IA amplitude. A low amplitude current would be expected to have a greater effect on a high-input-resistance cell (and *vice versa*), providing a potential explanation to the paradox noted by the Referee. In electrophysiological terms, the impact could be simply the degree of hyperpolarization and resonance frequency.